# EventCLIP: Adapting CLIP for Event-based Object Recognition

## Abstract

Recent advances in zero-shot and few-shot classification heavily rely on the success of pre-trained vision-language models (VLMs) such as CLIP. Due to a shortage of large-scale datasets, training such models for event camera data remains infeasible. Thus, adapting existing models across modalities is an important research challenge. In this work, we introduce EventCLIP, a novel approach that utilizes CLIP for zero-shot and few-shot event-based object recognition. We first generalize CLIP's image encoder to event data by converting raw events to 2D grid-based representations. To further enhance performance, we propose a feature adapter to aggregate temporal information over event frames and refine text embeddings to better align with the visual inputs. We evaluate EventCLIP on N-Caltech, N-Cars, and N-ImageNet datasets, achieving state-of-the-art few-shot performance. When fine-tuned on the entire dataset, our method outperforms all existing event classifiers. Moreover, we explore practical applications of EventCLIP including robust event classification and label-free event recognition, where our approach surpasses previous baselines designed specifically for these tasks.

## 1 Introduction

Event-based cameras have recently gained significant interest in computer vision due to their high temporal resolution, low energy consumption, and high dynamic range properties (Gallego et al., 2020). Event-based vision has shown promising results in various applications, such as object recognition (Gehrig et al., 2019; Kim et al., 2021), detection (Perot et al., 2020; Gehrig & Scaramuzza, 2022), tracking (Gallego et al., 2017; Gehrig et al., 2018; 2020b), and optical flow estimation (Zhu & Yuan, 2018; Gehrig et al., 2021). However, this novel imaging modality poses unique challenges, including the need for specialized models to handle the asynchronous nature of events, and the lack of large-scale datasets. As in classical recognition problems, newly captured event data can contain objects from categories that are not present in the training set of deployed models. In such cases, trained models will fail, and it may be infeasible to re-train the model every time a new object category is introduced, motivating the need for event-based zero-shot and few-shot recognition systems.

In frame-based vision, pre-trained vision-language models (VLMs) such as CLIP (Radford et al., 2021) have shown remarkable success in zero-shot and few-shot learning tasks. Trained on large-scale datasets, these models try to map paired images and texts to an aligned feature space. Open-world zero-shot classification is made possible by leveraging the feature similarity between unseen objects and texts with novel category names (Radford et al., 2021; Jia et al., 2021). Recently, several works have designed data-efficient methods to adapt CLIP under the few-shot learning setting for better accuracy (Gao et al., 2021; Zhou et al., 2022c;b; Zhang et al., 2022a; Wortsman et al., 2022). However, there is currently no large-scale event-text dataset available, making it impossible to train event-language models from scratch. This motivates us to ask the question: can 2D pre-trained VLMs be transferred to event-based vision and realize zero- or few-shot object recognition?

In this work, we propose EventCLIP as the first attempt introducing CLIP to event-based visual understanding. To bridge the gap between asynchronous event data and CLIP's frame-based input representation, we split an event stream into multiple time windows, and convert each into a 2D frame. Following Radford et al. (2021), text prompts are constructed by placing class names into hand-crafted templates, and text features are extracted as the zero-shot classifier weight. Each event frame is classified by CLIP individually, and the final result is obtained by simple voting.

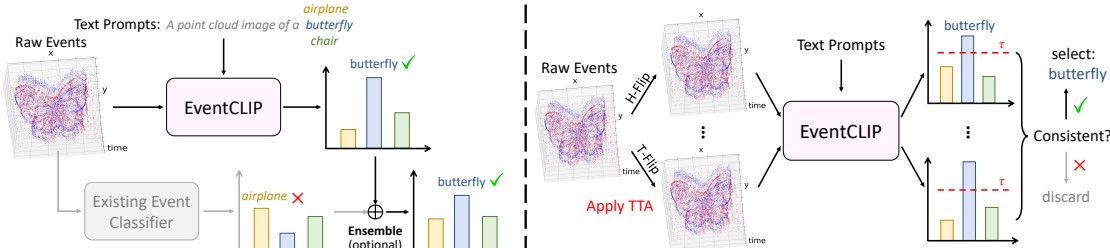

Figure 1: Existing event-based models are trained from scratch on closed-set datasets. They fail on events with unseen categories, camera motions or lighting. *Left*: our method adopts CLIP to perform zero-shot or few-shot open-world event recognition. Moreover, we can ensemble EventCLIP with existing event classifiers to enhance their robustness. *Right*: We leverage EventCLIP to learn from unlabeled data. To obtain reliable pseudo labels, we apply test-time-augmentation (TTA) such as horizontal and temporal flip to events, and only select examples with a consistent predicted category.

Although EventCLIP can achieve zero-shot classification, its performance still lags behind existing classifiers trained on event domain data. We thus propose to learn lightweight adapters to refine the pre-trained CLIP features. Different from previous work which only adapts one image feature (Gao et al., 2021) or a fixed number of features in pre-defined orders (Zhang et al., 2022b), the number and order of our event features depend on the camera trajectory. We thus design a Transformer-based adapter to aggregate temporal information from multiple frames. We also fine-tune text features as the weight of the output fully-connected layer in a classifier. With these careful designs, EventCLIP achieves more data-efficient few-shot learning compared to existing event-based classifiers.

Finally, we explore more applications of EventCLIP. We discover that the 2D pre-trained knowledge in CLIP complements the domain knowledge in models trained purely on event camera data. Therefore, we perform model ensemble using EventCLIP and existing event-based classifiers, which consistently improves their robustness against data corruption. In addition, since EventCLIP can classify unseen objects, we use it to create labels on unlabeled data for label-free learning. Leveraging the spatio-temporal property of events, we are able to train on high-quality pseudo labels.

In summary, this work makes four main contributions: **(i)** a novel framework leveraging pre-trained CLIP for event camera data, **(ii)** a Transformer-based feature adapter tailored to event temporal information aggregation, **(iii)** state-of-the-art few-shot and fine-tuning results on three datasets, and **(iv)** significantly improved robustness and unsupervised classification accuracy on N-ImageNet subsets.

## 2  RELATED WORK

In this section, we briefly review recent works in event-based classification, bridging frame-based and event-based vision, and CLIP-based transfer learning, which is further expanded in Appendix A.

**Deep Learning for Event-based Classification.** Depending on the utilization of the asynchronous nature of events, existing event-based classifiers can be mainly categorized into two classes, namely, synchronous and asynchronous methods. Synchronous models aggregate events to a grid-based representation, followed by standard modules such as Convolutional Neural Networks (CNNs) (Sironi et al., 2018; Gehrig et al., 2019; 2020a; Kim et al., 2021). Significant efforts have been made to achieve efficient and expressive event-to-frame conversion, such as binarized event occurrence (Cohen et al., 2018), event counts (Maqueda et al., 2018), and sorted event timestamps (Alzugaray & Chli, 2018). Recently, EST (Gehrig et al., 2019) has achieved state-of-the-art results with an end-to-end learnable event-to-frame conversion pipeline. To improve robustness against data noise, DiST (Kim et al., 2021) proposes to suppress noisy events leveraging their spatio-temporal relationships, which is proved effective under camera motion and lighting change during data capture. Compared to asynchronous methods, synchronous models achieve consistently better results across datasets (Gallego et al., 2020). As our primary goal is to achieve high accuracy instead of efficiency, we adopt representative synchronous event-based classifiers as our baselines in the experiments.

**Bridging Frame-based and Event-based Vision.** Inspired by the success of classical computer vision, several works have tried to bridge these two modalities. Some papers focus on reconstructing natural images from events, and then apply conventional deep models on the converted frames (Rebecq et al., 2019a; Stoffregen et al., 2020; Cho et al., 2023). However, they introduce computational

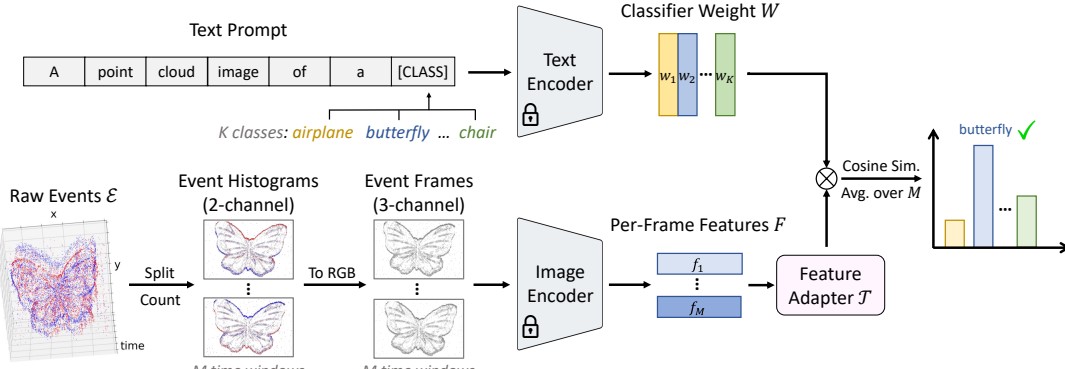

Figure 2: **EventCLIP overview**. Given raw events $\mathcal{E}$, we split it to $M$ time windows and construct event histograms. They are converted to RGB frames and fed to CLIP to obtain image features $F$. Meanwhile, we construct text prompts with $K$ class names, and extract text features as the classifier weight $W$. $F$ is updated with a feature adapter $\mathcal{T}$, and its cosine similarity with $W$ serves as the classification output. $\mathcal{T}$ is an identity function under the zero-shot setting, and a Transformer encoder in few-shot learning. The final result is obtained by averaging predictions from $M$ frames.

overhead which is at odds with event cameras' low-latency nature. Closer to ours are methods that transfer knowledge learned from images to event-based models (Hu et al., 2020; Wang et al., 2021a; Sun et al., 2022). However, they either require paired recordings of images and events, or massive labeled image data. In this work, we utilize CLIP pre-trained on RGB image-text pairs for data-efficient event-based classification. Our method converts events into frames via simple counting, and directly applies CLIP for zero-shot classification. We can further boost its performance via few-shot feature adaptation, without the need for paired RGB images or large amounts of labels.

**CLIP-based Few-Shot Transfer Learning.** Transfer learning aims to leverage models trained on large-scale datasets to facilitate learning in data-scarce scenarios. In event-based object recognition, existing methods also utilize models pre-trained on ImageNet (Deng et al., 2009) as their backbones to improve performance via fine-tuning (Maqueda et al., 2018; Gehrig et al., 2019; Kim et al., 2021). Trained on millions of image-text pairs, CLIP (Radford et al., 2021) learns transferable representations that are useful for downstream tasks. To further enhance its accuracy, CLIP-Adapter (Gao et al., 2021) and Tip-Adapter (Zhang et al., 2022a) learn lightweight modules over CLIP features.

Our work is inspired by PointCLIP (Zhang et al., 2022b), which projects point clouds to multi-view images for CLIP-based zero-shot and few-shot shape recognition. Different from PointCLIP, event data mostly capture the boundary information of objects, while point clouds depict complete object surfaces. In addition, we design a Transformer-based adapter for event temporal information fusion, while PointCLIP simply uses an MLP since their multi-view projections follow a pre-defined order.

## 3 METHOD

In Sec. 3.1, we briefly review CLIP for zero-shot image classification. Then, we introduce Event-CLIP which utilizes the pre-trained knowledge in CLIP for zero-shot event understanding (Sec. 3.2). When a few labeled examples are provided, we present EventCLIP with learnable feature adapters in Sec. 3.3. Finally, in Sec. 3.4, we apply EventCLIP to more tasks including robust event recognition and label-free event recognition. The overall pipeline of EventCLIP is illustrated in Fig. 2.

### 3.1 BACKGROUND: CLIP-BASED IMAGE CLASSIFICATION

The training objective of CLIP is to map images and texts to a joint embedding space. CLIP consists of two encoders for image and text inputs, respectively. During training, given a batch of image-text pairs, CLIP maximizes the cosine similarity between embeddings of positive pairs, while minimizing it for negative pairs using a contrastive loss. CLIP is trained on a collection of 400 million web-crawled image-text data. The large pre-training dataset enables the incorporation of diverse visual concepts, thereby enhancing the transferability of the learned features to downstream tasks.

As CLIP is trained to match image and text features, it naturally lends itself to zero-shot classification. Formally, let $f_x$ be the image feature extracted by CLIP from an image $x$. Meanwhile, we construct text prompts by placing category names into a pre-defined template such as "a photo

of a [CLASS]", and leverage CLIP to extract a set of $K$ text features $W = \{\boldsymbol{w}_i\}_{i=1}^{K}$, where $K$ denotes the number of classes. The probability of predicting class $i$ given $\boldsymbol{x}$ is then computed as:

$$p(y = i | \boldsymbol{x}) = \frac{\exp(\cos(\boldsymbol{w}_i, \boldsymbol{f_x})/\tau)}{\sum_{j=1}^{K} \exp(\cos(\boldsymbol{w}_j, \boldsymbol{f_x})/\tau)}, \tag{1}$$

where $\cos(\cdot, \cdot)$ denotes the cosine similarity between two vectors, and $\tau$ is a scaling factor learned by CLIP. The zero-shot inference does not require any in-domain training data, but can achieve competitive results with fully supervised baselines on 2D image datasets (Radford et al., 2021).

### 3.2 ZERO-SHOT EVENT UNDERSTANDING

Prior work in event recognition has shown that raw events can be converted to meaningful 2D frames such as edge maps (Gallego et al., 2020). This motivates us to prompt VLMs pre-trained in the RGB domain to process event camera data. We adopt CLIP due to its public availability and importance in the literature. However, the presented approach should generalize to other VLMs as well.

**Bridging the Modality Gap.** Event cameras record brightness changes at each sensing pixel, and output a sequence of events $\mathcal{E} = \{e_i = (x_i, y_i, t_i, p_i)\}$ each parameterized by its spatial position $(x_i, y_i)$, triggered timestamp $t_i$, and polarity $p_i \in \{-1, 1\}$. To obtain the grid-like representations CLIP requires, we convert raw events into a sequence of 2D frames. Specifically, we split an event stream $\mathcal{E}$ to $M$ time windows $\{\mathcal{E}_i\}_{i=1}^{M}$ by grouping every $N$ consecutive events. Grouping with event count ensures better robustness against camera speed compared to using fixed time intervals (Kim et al., 2021). For each $\mathcal{E}_i$, we construct a 2-channel histogram by counting the number of positive and negative events at each pixel. To obtain a 3-channel image, we first normalize the histogram to the range of $[0, 1]$, and then colorize it with a pre-defined RGB color map. Finally, we set the empty pixels to pure white for better visual quality following Klenk et al. (2022).

**Zero-Shot Classification.** After converting raw events to $M$ 2D frames, we leverage CLIP's image encoder to extract their features $F = \{\boldsymbol{f}_i\}_{i=1}^{M}$. We then construct text prompts with class names and hand-crafted templates, and use CLIP to extract text embeddings $W = \{\boldsymbol{w}_i\}_{i=1}^{K}$. The zero-shot prediction for each time window can be computed with Eq. 1. Here, the text template should reflect the domain-specific knowledge about event data, such as the visual property of the converted frames.

To obtain the final classification output, we need to aggregate predictions from $M$ time windows. PointCLIP (Zhang et al., 2022b) also faces this problem as they project a point cloud to multiple views. They simply assign hyper-parameters to weigh the importance of each view, which are fixed through the dataset. This is feasible since the 3D point clouds they considered are all aligned to a canonical pose, and thus the projected views for different data follow the same order (front, right, back, left, top, bottom). However, the temporal order of our time windows depends on the event camera's trajectory, which varies a lot across data samples, making a pre-defined set of weights sub-optimal. Inspired by DeepSets (Zaheer et al., 2017), we select the order-invariant set operation mean-pooling to average the classification probabilities from all time windows as the final output.

### 3.3 FEW-SHOT FEATURE ADAPTATION

With the event-to-frame conversion, we successfully transform a 2D CLIP into a zero-shot event classifier. However, zero-shot EventCLIP still underperforms domain-specifically trained classifiers. To close the accuracy gap, we consider the few-shot learning setting, where a few labeled examples are available per class. With limited data, it is impossible to fine-tune the entire CLIP model as it will lead to severe overfitting. Instead, we only refine the features extracted by a frozen CLIP model.

**Image Feature Adapter.** Our goal is to incorporate the event domain knowledge into the extracted image features $\{\boldsymbol{f}_i\}_{i=1}^{M}$ to obtain a refined representation $\{\boldsymbol{f}_i^{*}\}_{i=1}^{M}$. Prior works simply apply an MLP to update features since there is either only one feature vector per sample (image) (Gao et al., 2021; Wortsman et al., 2022), or the visual features follow a fixed order (point cloud) (Zhang et al., 2022b; Zhu et al., 2023). For example, PointCLIP concatenates the multi-view features into one vector and feeds it to the MLP-based adapter. Their performance will degrade significantly if we shuffle the order of projection views. In contrast, as discussed above, the final prediction of Event-CLIP should be order-invariant. Therefore, the output of our image feature adapter $\{\boldsymbol{f}_i^{*}\}_{i=1}^{M}$ should be *permutation-equivariant* to the input features $\{\boldsymbol{f}_i\}_{i=1}^{M}$. Also, we need an architecture that can

process an arbitrary number of inputs as $M$ varies across samples. Inspired by a recent CLIP-based video classifier (Ju et al., 2022), we apply a lightweight 2-layer Transformer encoder (Vaswani et al., 2017), $\mathcal{T}$, to aggregate the temporal information of event streams. To avoid overfitting, we employ residual connections from CLIP features to the Transformer output features $\{\tilde{\boldsymbol{f}}_i\}_{i=1}^{M}$:

$$\{\tilde{\boldsymbol{f}}_i\}_{i=1}^{M} = \mathcal{T}(\{\boldsymbol{f}_i\}_{i=1}^{M}), \ \ \boldsymbol{f}_i^* = \alpha\boldsymbol{f}_i + (1-\alpha)\tilde{\boldsymbol{f}}_i, \tag{2}$$

where $\alpha$ is a hyper-parameter controlling the ratio of original CLIP knowledge. After applying the visual adapter, we use the updated image features $\{\boldsymbol{f}_i^*\}_{i=1}^{M}$ and the text features $W$ to perform classification as done in the zero-shot setting. Thanks to the order invariance property, our few-shot EventCLIP is also more robust against different camera motions in the data capture process.

**Text Feature Adapter.** There have been several works studying data-efficient tuning of CLIP's text branch (Zhou et al., 2022c;b). As pointed out by He et al. (2023), all these methods aim at learning a better classifier weight $W = \{\boldsymbol{w}_i\}_{i=1}^{K}$. Therefore, we follow them to adopt the simple Classifier Tuning method (Wortsman et al., 2022), by fine-tuning $W$ with gradient descent. In our initial experiments, Classifier Tuning indeed achieves competitive performance with more complicated tuning methods such as prompt tuning (Zhou et al., 2022c;b), while requiring much less computation.

### 3.4 Extensions of EventCLIP

**Robust event classification.** Existing event-based classifiers are trained from scratch on event datasets. These datasets are often captured under limited environment variations (Kim et al., 2021). Therefore, the model performance degrades drastically when tested on unseen settings, such as changes in lighting or camera motion. On the contrary, CLIP is trained on large-scale datasets collected from the Internet, thus exhibiting high robustness against data corruption. A natural idea is to ensemble the two models for joint prediction. Specifically, we simply average the predicted logits from a pre-trained event-based classifier and a zero-shot or few-shot EventCLIP as the final output. As we will show in the experiments, the domain-specific event knowledge and the 2D pre-trained knowledge are able to complement each other, leading to state-of-the-art model robustness.

**Label-free event recognition.** In many practical scenarios, we have access to not only a few labeled events but also large amounts of unlabeled event data. The extreme setting in this line is label-free learning, where we only have raw events without any labels. In both cases, we can leverage the few-shot or zero-shot EventCLIP to create pseudo labels, and then fine-tune the model on them. To generate reliable pseudo labels, we run predictions on multiple augmented versions of the event, and only select data with a consistent predicted label. Formally, based on the fact that an event stream should remain the same class after horizontal flip and temporal reverse, given an event, we create four versions of it by applying the augmentations combinatorially. Then, we discard events with an inconsistent predicted class. To further improve the label quality, we employ a threshold $\tau$ to select high-confident samples, and only take the top-$k$ predictions per class for balanced model training.

## 4 Experiments

In this section, we evaluate EventCLIP on multiple settings. In Sec. 4.2, we study the best design choices to transfer CLIP's pre-trained knowledge to event camera data. Then, we show the performance gain from limited training data in few-shot learning (Sec. 4.3). When more data are available, EventCLIP can achieve state-of-the-art accuracy by fine-tuning the CLIP backbone (Sec. 4.4). In Sec. 4.5, we use the 2D pre-trained knowledge in CLIP to improve the robustness of existing event classifiers. Finally, in Sec. 4.6, we demonstrate learning from unlabeled events with our method.

### 4.1 Experimental Setup

**Datasets.** We use three publicly available datasets in our experiments: N-Caltech (Orchard et al., 2015), N-Cars (Sironi et al., 2018), and N-ImageNet (Kim et al., 2021). *N-Caltech* contains 8,246 samples from 101 classes, recorded by a moving $180 \times 240$ resolution ATIS system (Posch et al., 2010) in front of a monitor displaying still images from the original RGB Caltech101 dataset (Fei-Fei et al., 2004). In contrast, *N-Cars* provides event streams recorded by the ATIS system in a real-world urban environment. It contains 12,336 samples of the class car and 11,693 samples of the

Table 1: **Zero-shot EventCLIP and ablation experiments**. We report (a) zero-shot accuracy (%) of EventCLIP with our best design choice, followed by ablation studies on (b) the number of events per time window $N$, (c) the event histogram to RGB frame colorization method, (d) the pre-trained CLIP's image encoder, and (e) different templates for constructing the text prompt.

(a) Zero-shot accuracy (%)

| Dataset | N-Caltech | N-Cars | N-ImageNet |
|---|---|---|---|
| Acc. | 69.67 | 82.28 | 20.78 |

(b) Number of events $N$

| Dataset | N-Caltech | | | N-ImageNet | | |
|---|---|---|---|---|---|---|
| $N (\times 10^3)$ | 15 | 20 | 25 | 50 | 70 | 80 |
| 0-shot Acc. | **69.87** | 69.67 | 69.33 | 20.04 | **20.78** | 20.61 |
| 10-shot Acc. | 84.98 | **85.62** | 84.93 | 27.83 | **28.63** | 27.62 |

(c) Event to frame colorization

| Dataset | N-Caltech | | | N-Cars | | |
|---|---|---|---|---|---|---|
| Method | Gray | R-B | Learn | Gray | R-B | Learn |
| 0-shot Acc. | **69.67** | 65.93 | - | **82.28** | 78.39 | - |
| 10-shot Acc. | 85.62 | 82.87 | 85.69 | 84.77 | 83.26 | 84.85 |

(d) CLIP's Image encoder on N-Caltech

| Model | RN | RN×4 | RN×16 | RN×64 | ViT-B | ViT-L |
|---|---|---|---|---|---|---|
| Size (MB) | 244 | 403 | 631 | **1300** | 335 | 890 |
| 0-shot Acc. | 44.34 | 51.41 | 60.83 | 61.92 | 61.11 | **69.67** |
| 10-shot Acc. | 74.21 | 77.43 | 79.95 | 81.23 | 80.70 | **85.62** |

(e) Text prompt template on N-Caltech

| Prompt | 0-shot Acc. | 10-shot Acc. |
|---|---|---|
| A photo of a [CLASS] | 66.57 | 82.18 |
| An event camera photo of [CLASS] | 64.73 | 78.34 |
| An edge map of a [CLASS] | 68.70 | 84.07 |
| A sketch image of a [CLASS] | 69.64 | 85.16 |
| A point cloud image of a [CLASS] | 69.67 | **85.62** |
| [Learnable Tokens] + [CLASS] | - | 85.37 |

class background. Similar to N-Caltech, *N-ImageNet* is the event camera version of ImageNet (Deng et al., 2009). As the largest event camera dataset, it contains 1.78 million event streams and 1,000 classes. The data were captured with a moving $480 \times 640$ resolution Samsung DVS Gen3 event camera (Son et al., 2017). N-ImageNet also provides variants of the test set captured with different camera motions and brightness, serving as a benchmark to evaluate the robustness of event classifiers. See Appendix B for detailed descriptions of each variant. For few-shot training, we randomly sample a subset of data from each category. We always report the results on the entire test set.

**Baselines.** We compare EventCLIP with existing synchronous event-based classifiers, namely, EST (Gehrig et al., 2019), Event Histogram (Maqueda et al., 2018), Sorted Time Surface (Alzugaray & Chli, 2018), and DiST (Kim et al., 2021). See Appendix C for their implementation details. Notice that, we use ResNet34 (He et al., 2016) pre-trained on the RGB ImageNet (Deng et al., 2009) as the backbone for all the baselines following their original paper. For DiST and EST, we also tested larger backbones such as ResNet101 and ViT-L (Dosovitskiy et al., 2021), but did not observe clear improvement as will be shown later. We will introduce other baselines in each task below.

**Our Implementation Details.** To convert events into frames, we set the number of events per time window $N$ as 20,000, 10,000, and 70,000 on N-Caltech, N-Cars, and N-ImageNet, respectively. This accounts for each dataset's event camera resolution. For colorizing the events, i.e., converting the 2-channel event histograms to 3-channel RGB images, we simply use a gray-scale color map by multiplying both positive and negative event counts with [127, 127, 127]. We use the ViT-L/14 image encoder in CLIP. We select "`a point cloud image of a [CLASS]`" as the text template.

## 4.2 ZERO-SHOT CLASSIFICATION

**Results.** Table 1(a) shows the zero-shot classification accuracy of EventCLIP. Without any in-domain training, our method achieves an accuracy of 69.67% on N-Caltech which has 101 classes. This proves the effectiveness of our event-to-frame conversion pipeline in bridging the RGB and event camera domains. In addition, our model scores a higher 82.28% accuracy on N-Cars which is captured in the real world, demonstrating its generalizability. On the challenging N-ImageNet dataset, we achieve a lower accuracy of 20.78% due to the lack of event domain knowledge.

**Ablation Study.** We first study the event time window size $N$ in Table 1(b). Since event streams from N-Cars are generally sparse, we convert all events to one frame without ablation. On N-Caltech, a smaller $N$ achieves the best zero-shot accuracy, but we select 20,000 since it strikes a better balance between zero- and few-shot results. We need a higher $N = 70,000$ for optimal performance on N-ImageNet as its larger camera resolution triggers more events. See Appendix Fig. 6 for more ablation experiments on $N$. Table 1(c) ablates different ways of colorizing the event histogram. We test the red-blue color map (dubbed R-B) commonly used to visualize events, which multiplies positive and negative event counts with [255, 0, 0] and [0, 0, 255], respectively. It leads to much worse results because the color statistics of their converted images are distinct from the natural images CLIP is trained on. We also design a learnable method by initializing the color map with

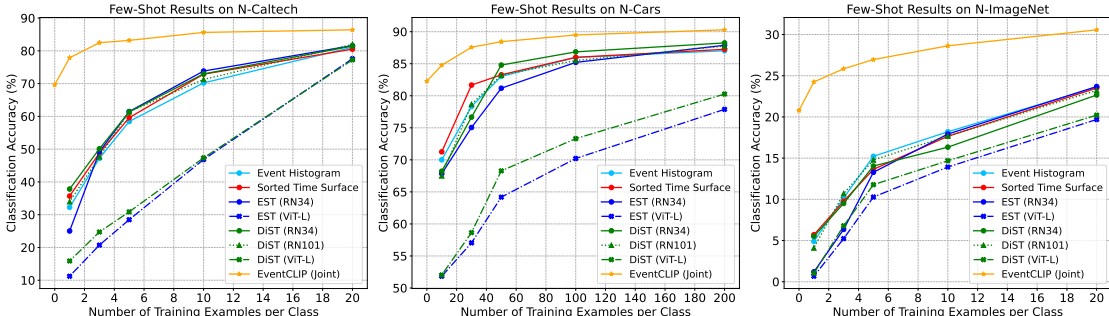

Figure 3: **Few-shot classification accuracy** on N-Caltech, N-Cars, and N-ImageNet. All baselines build upon the ResNet34 backbone pre-trained on the RGB ImageNet dataset. For EST and DiST, we try additional variants with larger ResNet101 and ViT-L backbones. Our EventCLIP consistently outperforms state-of-the-art event-based classifiers across all numbers of shots on three datasets.

two vectors and optimizing them with gradient descent. This leads to slightly better accuracy, but requires $10\times$ the computation since it needs to backpropagate through the heavy ViT image encoder. Overall, our gray-scale color map is efficient and reduces the input's domain gap to CLIP.

One important property of large-scale pre-trained models is the scalability of their performance with model size. We study this effect using different image encoders of CLIP in Table 1(d). Within the ResNet (He et al., 2016) family and the ViT (Dosovitskiy et al., 2021) family, EventCLIP achieves higher zero- and few-shot accuracy as the model size grows. Notably, ViT-L significantly outperforms RN50×64 in both accuracy, despite having much fewer parameters. The reason might be the converted event frames mostly capture the object boundary, and are thus biased towards the shape information. Studies have shown that CNNs are usually texture-biased (Geirhos et al., 2019), while ViTs are better at processing shape information (Naseer et al., 2021). This observation may serve as future guidelines in designing event-based vision algorithms. Finally, in Table 1(e), we study the text prompt used to generate the classifier weight $W$. The prompt "A photo of a [CLASS]" commonly used in 2D vision tasks achieves $66.57\%$ zero-shot accuracy. Simply prefixing "photo" with "event camera" leads to worse results as CLIP is not trained on event data. Instead, we explicitly describe the visual property of the event frames. Since events are mostly triggered by object boundaries, "edge map" and "sketch image" both lead to better results. Surprisingly, describing event frames with "point cloud image" achieves the highest accuracy, which aligns with previous works that treat raw events as spatio-temporal points (Wang et al., 2019). In addition, we tried prompt tuning with learnable textual tokens (Zhou et al., 2022c), which achieves similar few-shot performance. However, it trains $5\times$ slower as it requires backpropagation through the text encoder.

### 4.3 FEW-SHOT CLASSIFICATION

**Settings.** We experiment with 1, 3, 5, 10, 20 shots on N-Caltech and N-ImageNet. Since N-Cars only has two categories, we multiply the number of shots by 10. We test EventCLIP with four variants of adapters: (i) PointCLIP's MLP-based visual adapter (Zhang et al., 2022b), (ii) our proposed Transformer-based visual adapter, (iii) the Classifier Tuning text adapter (Wortsman et al., 2022), and (iv) a joint adapter combining (ii) and (iii). See Appendix C for our implementation details.

**Results.** We first compare EventCLIP using joint feature adapter with baselines in Fig. 3. On each dataset, all baselines with the ResNet34 backbone achieve similar performance, which aligns with previous work (Kim et al., 2021). In contrast, EventCLIP with feature adaptation achieves significantly higher few-shot accuracy. Our 20-shot accuracy on N-Caltech ($85.62\%$) even surpasses the best-performing model trained on the entire N-Caltech (EST's $81.7\%$) by around $4\%$. Notice that, all baselines are initialized with backbones pre-trained on RGB ImageNet, which is the source for creating N-ImageNet. Still, EventCLIP achieves consistently higher accuracy across all numbers of shots. Overall, the results prove that CLIP's large-scale pre-training learns generalizable representations, enabling quick adaptation to the new event camera domain with limited training data.

For a fair comparison, we also trained baselines with larger ImageNet pre-trained backbones ResNet101 and ViT-L. Naively fine-tuning ViTs on limited data usually leads to severe overfitting (Dosovitskiy et al., 2021), so we adopt the state-of-the-art data-efficient ViT training strategy (Touvron et al., 2021). As shown in the figures, DiST with ResNet101 leads to similar performance, while ViT-L results in much worse accuracy even with the advanced training strategy. This indicates the superiority of our EventCLIP framework as it scales well with larger models.

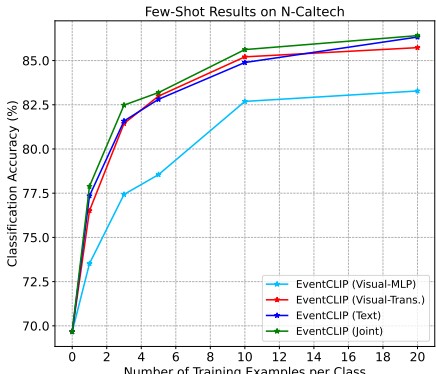

Figure 4: **Different feature adapters** of EventCLIP on N-Caltech. Trans. stands for the Transformer-based visual adapter.

Table 2: **Fine-tuning results**. We compare EventCLIP with E-CLIP (Zhou et al., 2023) which fine-tunes the CLIP image encoder. We report results under the few-shot setting or trained on all data in the dataset (All).

(a) Fine-tuning results on N-Caltech

| Data per Class | 1 | 2 | 5 | 10 | 20 | All |
|---|---|---|---|---|---|---|
| E-CLIP | 66.72 | 75.87 | 82.35 | 86.92 | 90.51 | 93.89 |
| **EventCLIP** | **75.82** | **78.86** | **83.57** | 87.42 | 90.41 | 93.57 |

(b) Fine-tuning results on N-ImageNet

| Data per Class | 1 | 2 | 5 | 10 | 20 | All |
|---|---|---|---|---|---|---|
| E-CLIP | 22.22 | 26.85 | 28.70 | 30.56 | 35.11 | 51.85 |
| **EventCLIP** | **24.39** | **27.23** | **31.12** | **34.24** | **38.28** | **53.20** |

**Ablation Study.** Figure 4 compares the performance of EventCLIP with four different feature adapters on N-Caltech. The Transformer-based visual adapter consistently outperforms the MLP-based counterpart by a sizeable margin, showing the importance of its permutation-equivariant property. Besides, using only the image feature adapter yields comparable results to using only the text feature adapter. Moreover, EventCLIP with joint feature adapter is slightly better than single-branch adapters with fewer than 10-shots data, and the performance is similar under 20-shots. Such observation is in line with previous work on RGB vision tasks (Gao et al., 2021). In our experiments, we observe more severe overfitting using joint adapters, as their validation loss generally increases faster than single-branch adapters. See Appendix D.1 for more ablations on N-Cars and N-ImageNet.

## 4.4 FINE-TUNING EVENTCLIP

**Settings.** Few-shot feature adaptation is able to improve the model accuracy with minor computation and time overhead. However, it still underperforms event-based classifiers trained on the entire dataset. To show that EventCLIP can also achieve state-of-the-art performance when more data are available, we propose to fine-tune the CLIP's image encoder jointly with the text feature adapter.

**Baseline.** We compare with a concurrent work E-CLIP (Zhou et al., 2023), which re-trains CLIP's image encoder to obtain an event-specific encoder. With full training data, E-CLIP outperforms all previous event-based classifiers. Different from us, E-CLIP requires *paired RGB images* with events during training. For a fair comparison, we adopt CLIP with the same ViT-B/16 backbone as they do.

**Results.** We compare our fine-tuning results with E-CLIP in Table 2. Following their protocol, we also report the accuracy in the few-shot setting. On N-Caltech, we achieve better results in the low data regime. With more data, EventCLIP is competitive with E-CLIP. We hypothesize that this is because N-Caltech is extremely class-imbalanced where some categories only have 20 events. Thus, the paired RGB images provide much useful information for E-CLIP. This is verified by the results on N-ImageNet with abundant samples per class. Without using additional data, EventCLIP still outperforms E-CLIP consistently over all settings, achieving new state-of-the-art performance.

## 4.5 ROBUST EVENT CLASSIFICATION

**Settings.** Our goal is to evaluate whether the large-scale 2D pre-trained knowledge in CLIP is complementary to existing event-based classifiers. Therefore, we perform model ensemble using 0-, 20-, and 100-shot EventCLIP (few-shot trained with the joint feature adapter) with baselines trained on *the entire N-ImageNet* by simply averaging the predicted class logits. We test the original and the ensembled model on the normal test set of N-ImageNet, as well as its 9 variants (Kim et al., 2021) which are captured under out-of-distribution camera motions and lighting conditions.

**Baseline.** We adopt pre-trained weights of baselines from the official codebase of N-ImageNet. We exclude EST here since there is no pre-trained weight available. As an ensemble baseline, we report the accuracy of model ensemble with DiST, as DiST is currently the most robust event-based classifier. We also compare with Ev-TTA (Kim et al., 2022), which is designed specifically for event domain adaptation. Importantly, Ev-TTA requires access to *data in the new domain* to perform test-time adaptation to update model weights. It has an online version where novel testing data are used only once and an offline version where the out-of-distribution data are used multiple times.

Table 3: **Classification accuracy on N-ImageNet and its robustness variants**. The numbers of baseline event-based classifiers and Ev-TTA are copied from Kim et al. (2022). Average is computed over 9 variants. Best results are bold and the second-best results are underlined.

| Data Variation | None | Trajectory | | | | | Brightness | | | | Average |
|---|---|---|---|---|---|---|---|---|---|---|---|
| Variant ID. | Orig. | 1 | 2 | 3 | 4 | 5 | 6 | 7 | 8 | 9 | All |
| **Event Histogram** | 47.73 | 43.73 | 33.72 | 37.69 | 24.56 | 35.24 | 20.89 | 29.68 | 36.33 | 34.56 | 32.93 |
| + DiST | 51.67 | 48.02 | 38.18 | 43.16 | 27.56 | 40.02 | 25.19 | 34.22 | 40.63 | 38.83 | 37.31 |
| + Ev-TTA (online) | - | 44.94 | 44.63 | 43.31 | 41.48 | 43.46 | 26.89 | 34.71 | 43.86 | 43.42 | 40.86 |
| + Ev-TTA (offline) | - | 48.64 | 48.01 | **47.24** | 44.49 | **47.06** | 30.08 | 38.34 | 47.37 | 46.58 | 44.20 |
| + EventCLIP (0-shot) | 50.03 | 48.49 | 43.33 | 41.57 | 37.90 | 40.14 | 25.72 | 34.28 | 44.33 | 44.65 | 40.05 |
| + EventCLIP (20-shot) | 51.68 | 51.06 | 46.58 | 43.63 | 42.59 | 42.94 | 27.64 | 37.18 | 47.65 | 46.93 | 42.91 |
| + EventCLIP (100-shot) | **52.72** | **51.98** | **48.82** | 44.98 | **44.75** | 44.79 | 30.26 | **38.84** | **49.68** | **48.20** | **44.70** |
| **Sorted Time Surface** | 47.90 | 44.33 | 33.50 | 40.17 | 23.72 | 37.19 | 21.57 | 30.31 | 36.63 | 35.18 | 33.62 |
| + DiST | 51.56 | 47.92 | 37.92 | 43.84 | 27.07 | 40.64 | 25.38 | 34.35 | 40.49 | 38.87 | 37.39 |
| + Ev-TTA (online) | - | 46.02 | 45.29 | 45.91 | 42.53 | 43.90 | 26.70 | 36.17 | 45.00 | 45.22 | 41.86 |
| + Ev-TTA (offline) | - | 49.58 | 47.67 | **48.36** | 45.59 | **46.72** | 30.07 | 39.30 | 48.24 | 47.76 | 44.81 |
| + EventCLIP (0-shot) | 50.25 | 48.78 | 43.12 | 44.03 | 37.24 | 41.66 | 26.23 | 35.21 | 44.39 | 43.82 | 40.50 |
| + EventCLIP (20-shot) | 52.23 | 51.27 | 46.65 | 46.07 | 42.38 | 44.22 | 28.22 | 37.95 | 48.04 | 47.22 | 43.56 |
| + EventCLIP (100-shot) | **52.85** | **52.78** | **48.92** | 47.44 | 44.68 | 45.55 | 30.53 | 39.68 | **49.63** | **49.19** | 45.38 |
| **DiST** | 48.43 | 45.17 | 36.58 | 42.28 | 26.57 | 38.70 | 24.39 | 32.76 | 38.99 | 37.37 | 35.89 |
| + Ev-TTA (online) | - | 46.32 | 46.05 | 46.57 | 43.23 | 44.58 | 28.05 | 36.98 | 46.03 | 45.64 | 42.61 |
| + Ev-TTA (offline) | - | 48.53 | 47.75 | **48.38** | 45.35 | **47.26** | 31.02 | 39.07 | 48.19 | 47.66 | 44.80 |
| + EventCLIP (0-shot) | 50.53 | 49.36 | 44.47 | 45.12 | 37.83 | 43.15 | 28.01 | 36.79 | 45.72 | 44.58 | 41.67 |
| + EventCLIP (20-shot) | 52.28 | 51.42 | 47.53 | 46.77 | 42.34 | 45.05 | 29.62 | 39.05 | 49.00 | 47.52 | 44.26 |
| + EventCLIP (100-shot) | **53.12** | **52.45** | **49.01** | 47.78 | 43.85 | 46.22 | 31.00 | **39.92** | **49.77** | **48.74** | **45.42** |

**Results.** As shown in Table 3, model ensemble with 0-shot EventCLIP already increases the accuracy of baselines by more than 5%, which is higher than ensemble with a fully trained DiST. This indicates that pre-trained CLIP contains information that cannot be effectively learned from event camera datasets only. Such information complements pre-trained event-based classifiers, making them more robust against data corruption. Compared to ensemble with DiST, ensemble with 20-shot EventCLIP improves the average accuracy on robustness variants by more than 5%. It is worth noting that DiST achieves 48.43% accuracy on the original test set, while our 20-shot EventCLIP scores a much lower 30.57%. Besides, ensemble DiST with a worse performing EventCLIP still greatly improves the model performance on both the original and the robustness variants of N-ImageNet.
Next, we compare our method with the test-time adaptation method Ev-TTA, which requires access to additional out-of-distribution data. Surprisingly, EventCLIP trained on 20 samples per category (less than 2% of all training data) outperforms the online version of Ev-TTA on most of the subsets. With 8% of training data, our 100-shot EventCLIP outperforms the offline version of Ev-TTA, achieving new state-of-the-art robustness results. See Appendix D.5 for qualitative analysis.

### 4.6 Label-Free Event Recognition

**Settings and Baseline.** We follow the protocol of Ev-LaFOR (Cho et al., 2023) to perform *fully unsupervised learning* on a 100-class subset of N-ImageNet. Please refer to their paper for details on the N-ImageNet (Mini) dataset. Ev-LaFOR reconstructs images from events and uses CLIP for classification. For a fair comparison, we adopt CLIP with the same ViT-B/32 backbone as they do.

**Results.** EventCLIP with ViT-B/32 backbone achieves a zero-shot accuracy of 27.08% on the test set of N-ImageNet (Mini). With our carefully designed pseudo-label creation process, self-training further boosts the performance to 35.26%, surpassing Ev-LaFOR's accuracy of 31.28% by around 4%. See Appendix D.3 for additional results under the semi-supervised learning setting.

## 5 Conclusion

In this paper, we propose EventCLIP, which performs cross-modality event recognition with CLIP. With event-to-frame conversion, we successfully transfer CLIP's 2D pre-trained knowledge to the event camera domain. To further enhance the performance, we develop lightweight adapters to refine the pre-trained CLIP embeddings. Moreover, EventCLIP can be employed to improve the robustness of existing classifiers via model ensemble, or learn from unlabeled data with self-training. Our work opens up new possibilities to apply recent advances in vision foundation models to event-based vision. We discuss the limitations and potential future directions of this work in Appendix E.

ETHICS STATEMENT

We do not see significant risks of human rights violations or security threats in our work. However, since our method builds upon CLIP pre-trained on large-scale datasets, it inevitably inherits the bias in training data. Future research should also avoid malicious uses such as social surveillance, and be careful about the training cost of large vision foundation models. Overall, the technical outcomes of this paper need to cooperate with humans to avoid negative societal impacts.

REPRODUCIBILITY STATEMENT

All of our methods are implemented in PyTorch (Paszke et al., 2019) and can be trained with 1 modern GPU in less than 1 day, enabling both industrial and academic researchers. To ensure the reproducibility of our work, we provide detailed descriptions of the implementation details and hyper-parameters of baselines and our method in Appendix C. To facilitate future research, we will release the code of our work and the pre-trained model weights upon acceptance of this paper.

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

## A ADDITIONAL RELATED WORK

**Deep Learning for Event-based Classification.** Depending on the utilization of the sparsity and asynchronous nature of event data, existing event-based classifiers can be mainly categorized into two classes (Gallego et al., 2020), namely, synchronous and asynchronous methods. Synchronous models aggregate events to a grid-based representation, and then use standard modules such as Convolutional Neural Networks (CNNs) to process it (Alzugaray & Chli, 2018; Maqueda et al., 2018; Sironi et al., 2018; Gehrig et al., 2019; 2020a; Hu et al., 2020; Kim et al., 2021). Significant efforts have been made to achieve efficient and expressive event-to-frame conversion (Cohen et al., 2018; Maqueda et al., 2018; Alzugaray & Chli, 2018). Recently, EST (Gehrig et al., 2019) has achieved state-of-the-art results in classification with an end-to-end learnable event-to-frame conversion pipeline. As a remedy for robustness in the presence of noise, DiST (Kim et al., 2021) proposes to suppress noisy events leveraging their spatio-temporal relationships, which is proved effective under camera motion and lighting variations.

On the other hand, asynchronous networks (Lee et al., 2016; Neil et al., 2016; Amir et al., 2017; Lagorce et al., 2016; Messikommer et al., 2020; Cannici et al., 2020; Li et al., 2021; Schaefer et al., 2022) have been developed to address the computational latency inherent in grid-based methods, which directly apply Spiking Neural Networks or Graph Neural Networks to raw event inputs. However, these methods still consistently underperform synchronous methods across datasets (Orchard et al., 2015; Sironi et al., 2018; Kim et al., 2021). As our primary goal is to achieve high accuracy instead of efficiency, we adopt the former category of methods as our baselines in the experiments.

**Bridging Frame-based and Event-based Vision.** Inspired by the success of classical computer vision, several works have introduced techniques from frame-based vision to process event data. Some papers focus on reconstructing natural images from events, and then apply conventional deep models on the converted frames (Rebecq et al., 2019a;b; Scheerlinck et al., 2020; Stoffregen et al., 2020; Weng et al., 2021; Cho et al., 2023). However, they introduce computational overhead which is at odds with event cameras' low-latency nature. Another line of work tries to simulate event data from existing RGB image datasets, where ground-truth annotations can be automatically obtained from labeled frames (Mueggler et al., 2017; Rebecq et al., 2018; Hu et al., 2021b; Gehrig et al., 2020a; Zhu et al., 2021). The drawback is the large Sim2Real gap of the synthesized events such as unrealistic camera motions and the absence of sensor noises. The most relevant works to ours are methods that transfer knowledge learned from RGB images to event-based models (Hu et al., 2020; Wang et al., 2021a; Sun et al., 2022; Zanardi et al., 2019a;b; Messikommer et al., 2022). However, they either require paired recordings of image and event data, or massive labels in the image domain. In this work, we utilize CLIP pre-trained on RGB image-text pairs for data-efficient event-based classification. Our method converts events into frames via simple counting, and directly applies CLIP for zero-shot classification. We can further boost its performance via few-shot fine-tuning, without the need for paired RGB images or large amounts of labels.

**CLIP-based Few-Shot Transfer Learning.** Transfer learning aims to leverage models trained on large-scale datasets to help learning on data-scarce tasks. In the field of event-based object recognition, existing grid-based methods have also utilized models pre-trained on RGB images from ImageNet as their backbones to improve performance via fine-tuning (Gehrig et al., 2019; Kim et al., 2021; Maqueda et al., 2018; Alzugaray & Chli, 2018). Trained on large-scale image-text pairs, CLIP (Radford et al., 2021) has shown great potential in learning transferable representations for various downstream tasks. To further enhance the few-shot classification accuracy of CLIP, CoOp (Zhou et al., 2022c), and CoCoOp (Zhou et al., 2022b) insert learnable text tokens to perform task-specific prompt tuning. CLIP-Adapter (Gao et al., 2021), Tip-Adapter (Zhang et al., 2022a), and WiSE-FT (Wortsman et al., 2022) instead learn lightweight adapters over CLIP features.

In addition to 2D image classification, CLIP has also been extended to 2D detection (Gu et al., 2022; Zhou et al., 2022d), segmentation (Rao et al., 2022; Zhou et al., 2022a), and video analysis (Wang et al., 2021b; Ju et al., 2022). Our work is inspired by PointCLIP (Zhang et al., 2022b; Zhu et al., 2023), which projects point clouds to multi-view images and performs zero-shot and few-shot shape recognition with CLIP. Different from PointCLIP, events only capture the boundary information of objects compared to the complete surfaces presented in point clouds. Also, we design a Transformer-based adapter for event temporal information fusion, while PointCLIP simply uses an MLP since their multi-view projections follow a fixed order.

## B    DETAILS ON N-IMAGENET ROBUSTNESS VARIANTS

Here, we provide more information about the robustness variants of N-ImageNet (Kim et al., 2021) test sets. The original training and testing set (it is actually the validation set, but we call it test set for simplicity) of N-ImageNet are both captured with a Samsung DVS Gen3 (Son et al., 2017) event camera moving around the screen under the same environmental conditions. To test event-based classifiers' robustness against variations in the data capture process, the authors create 9 variants of the same test set. Variants 1-5 change the camera motions used to trigger events, where different moving directions, frequencies, and amplitudes of the camera trajectory are employed. Variants 6-9 alternate the lighting conditions of the environment, such as extremely low or high illuminations. Overall, these variations cause a large degradation in the performance of existing event-based classifiers. See Appendix D.5 for visualizations of some data variants.

## C    IMPLEMENTATION DETAILS

In our few-shot experiments, all models are trained on the same subset of data. All reported results are averaged over three runs, and we empirically find that the performance variation is small.

**Baselines.** *EST* (Gehrig et al., 2019) is the state-of-the-art method on N-ImageNet which utilizes learnable kernels to convert raw events into grid-based representations. *Event Histogram* (Maqueda et al., 2018) converts the event counts into a two-channel image grouped by their polarity. *Sorted Time Surface* (Alzugaray & Chli, 2018) adopts the sorted indices of event timestamps to ensure durability against camera speed changes. *DiST* (Kim et al., 2021) is specifically designed to improve the robustness against event camera noise and motion variations. It achieves state-of-the-art results on the robustness benchmark of N-ImageNet variants.

We adopt the online official implementation of EST[1] (Gehrig et al., 2019) and DiST[2] (Kim et al., 2021). The implementation of Event Histogram and Sorted Time Surface are also adopted from the DiST codebase. We re-train all the models with their default settings on each dataset, but decrease the learning rate and number of training epochs when observing severe overfitting. ResNet34 (He et al., 2016) pre-trained on the RGB version of ImageNet (Deng et al., 2009) is adopted as their backbones, and fine-tuned jointly under the few-shot setting. As shown in the experiments, we tried DiST with pre-trained ResNet50 and ResNet101 backbones, but did not observe clear improvement in accuracy. We also tried pre-trained ViT (Dosovitskiy et al., 2021) models. Even with state-of-the-art data-efficient training strategy (Touvron et al., 2021), the joint fine-tuning suffers from severe overfitting, leading to results even worse than ResNet backbones. We hypothesize that this is because vision transformers are data-hungry, and thus are not suitable for the few-shot learning setting. To evaluate the model ensemble performance on the robustness variants of N-ImageNet, we directly use the pre-trained weights from the official release.

**Data Augmentation.** Following DiST (Kim et al., 2021), we use random jittering, random horizontal flip along the spatial dimension, and random reverse along the temporal dimension as data augmentations. We tried other event augmentations such as random event dropping, and random cropping over both spatial and temporal dimensions, but did not observe clear improvement. Since we convert events to 3-channel RGB frames, EventCLIP additionally benefits from the well-studied RGB image augmentation literature. We apply RandAugment (Cubuk et al., 2020) [3] to the converted event frames during training. Note that, for each loaded event stream, we apply the same set of operations to all frames converted from this data. RandAugment consistently improves the performance of EventCLIP on N-Caltech and N-ImageNet, while bringing little gain on N-Cars. Finally, the resulting frames are resized and center cropped to $224 \times 224$ following CLIP paper.

**Few-shot EventCLIP.** We adopt the pre-trained weights of CLIP from their official online release[4]. For EventCLIP with the Transformer-based visual adapter, we stack 2 standard Transformer encoder modules (Vaswani et al., 2017), with a token size equal to 256 and 4 heads. We choose the pre-LN Transformer variant (Xiong et al., 2020) as it leads to more stable training and less overfitting. To

---

[1]https://github.com/uzh-rpg/rpg_event_representation_learning

[2]https://github.com/82magnolia/n_imagenet

[3]We use the implementation from torchvision

[4]https://github.com/openai/CLIP

ensure the permutation-equivariant property, we do not apply positional encoding to the Transformer input following Yin et al. (2022) and Wu et al. (2023). For the MLP-based visual adapter baseline, we adopt the best-performing setting from PointCLIP (Zhang et al., 2022b), which concatenates image features $F = \{f_i\}_{i=1}^{M}$ to extract a global feature, and fuses with per-frame features via residual connections. For the text adapter, we treat the text features $W = \{w_i\}_{i=1}^{K}$ as the weight of the fully-connected layer in a classifier, and update it via gradient descent. When applying the visual adapter only, we set the residual ratio $\alpha$ to $0.5$, while we use $\alpha = 0.8$ to further alleviate overfitting when training two adapters jointly. On N-ImageNet, we always use $\alpha = 0.95$.

We train all models with the Adam (Kingma & Ba, 2014) optimizer for 100 epochs on N-Caltech and N-ImageNet, and 50 epochs on N-Cars. We use a batch size of 32 on N-Caltech and N-Cars, and 128 on N-ImageNet. When the number of training data is smaller than 32, e.g. N-Cars under the 10-shot learning setting only has 20 samples, we set the batch size as the number of data available. On N-Caltech and N-Cars, when applying the visual and text adapter separately, we set the peak learning rate as $2 \times 10^{-4}$ and $1 \times 10^{-3}$ for them, respectively. When training them jointly, we set the peak learning rate as $2 \times 10^{-4}$. Besides, we divide the learning rate by a factor of 10 when training on N-ImageNet to present overfitting. We also adopt a linear learning rate warmup schedule during the first $5\%$ of training steps, and decay the learning rate to 0 in a cosine schedule. We do not use any weight decay or gradient clipping as we did not find them useful in preliminary experiments.

**Fine-tuning EventCLIP.** We fine-tune the image encoder of CLIP, while keeping the text encoder frozen. We use the same hyper-parameters and training settings as few-shot EventCLIP, except that we use a $10\times$ smaller learning rate on CLIP's image encoder. When fine-tuned on the entire dataset, we train shorter for 50 epochs. Besides tuning all the model parameters, we also tried other parameter-efficient fine-tuning methods such as only tuning the bias terms, only tuning LayerNorm, and LoRA (Hu et al., 2021a), but observed worse performance compared to naive fine-tuning.

**Robust event classification with EventCLIP.** We take the zero-shot and few-shot EventCLIP and average the class logits with class logits predicted by baseline event-based classifiers. We search for the weight factors to balance the two terms on one subset, and fix it for all the remaining subsets.

**Learning from unlabeled data with EventCLIP.** There are two settings in this task: fully unsupervised learning where no labels are available, and semi-supervised learning where we have a few labeled samples per class. In the unsupervised setting, we use the zero-shot EventCLIP to generate pseudo labels. In the semi-supervised setting, we first train EventCLIP on the labeled data similar to few-shot learning, and then use it to generate pseudo labels. We choose horizontal flip and temporal flip as TTA methods. Given an unlabeled event $\mathcal{E}$, we first generate four versions of it $\{\mathcal{E}_i\}_{i=1}^{4}$ by applying the augmentations combinatorially. Then, we run EventCLIP to predict class probabilities $\{p_i \in \mathbb{R}^K\}_{i=1}^{4}$ ($K$ is the number of classes), and get the class labels $\{c_i = \arg\max_c p_i\}_{i=1}^{4}$. We discard examples with an inconsistent $\{c_i\}_{i=1}^{4}$. To further enhance the label quality, we select examples with confidence scores higher than $\tau$, i.e., $\max(p_i) > \tau, \forall i \in \{1, 2, 3, 4\}$. Finally, we take the top-$k$ most confident examples from each category to form the pseudo-labeled training set. After obtaining the training data, we follow few-shot EventCLIP to train our model on it.

For hyper-parameters, we choose $k = 30$ in both settings. For $\tau$, we need to use a very high value of $0.999$ in unsupervised learning, as the zero-shot EventCLIP is over-confident in its predictions. After few-shot adaptation, EventCLIP is better calibrated, and thus we can use a lower value of $0.5$.

## D  ADDITIONAL EXPERIMENTAL RESULTS

### D.1  ABLATION STUDY ON FEATURE ADAPTERS

We perform ablation studies on the feature adapters on N-Cars and N-ImageNet. On N-Cars, we test an additional text adapter with prompt tuning (dubbed as CoOp (Zhou et al., 2022c)), which learns context vectors in text prompts instead of using pre-defined templates. Fig. 5 presents the few-shot accuracy of different adapters. CoOp adapter is consistently worse than other adapters, while consuming much more resources as it requires backpropagation through CLIP's text encoder. Similar to N-Caltech, the joint feature adapter performs better in the low-shot scenarios, and the gap becomes negligible when more data are provided. On N-ImageNet, the joint adapter is clearly better than both single-branch adapters. Overall, we recommend users choose the text feature adapter as

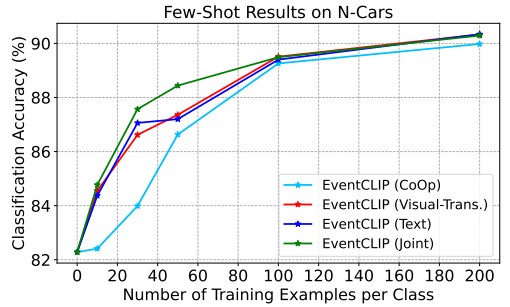 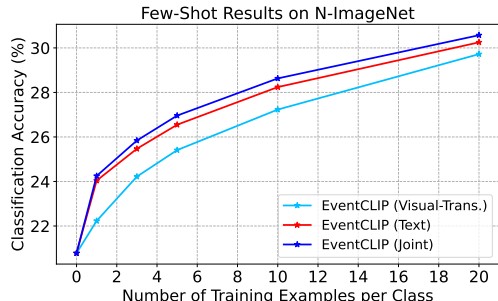

Figure 5: **Ablation on feature adapters** on N-Cars and N-ImageNet. Trans. stands for the Transformer-based visual adapter. CoOp means the text adapter based on prompt tuning.

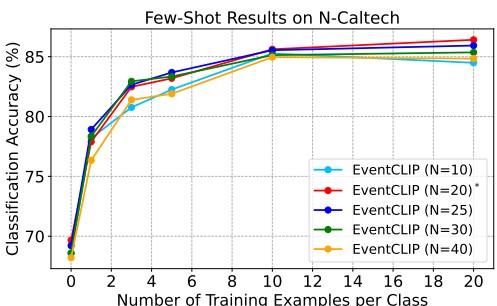 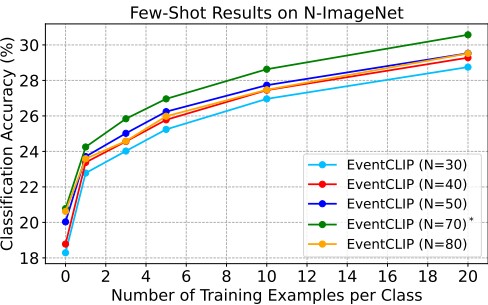

Figure 6: **Ablation on event time window size** $N$ ($\times 10^3$). Entry with $^*$ is the optimal setting.

Table 4: **Semi-supervised learning results**. We report the accuracy of EventCLIP trained only on few-shot labeled data vs. few-shot labeled data plus generated pseudo-labels.

| Labeled Data per Class | 1 | 3 | 5 | 10 | 20 |
|---|---|---|---|---|---|
| Few-shot only | 30.54 | 32.64 | 32.92 | 37.04 | 39.58 |
| Few-shot + Pseudo-labels | **35.98** | **37.04** | **37.53** | **40.30** | **42.02** |

a starting point, since it achieves competitive performance, while requiring less computation and hyper-parameter tuning. Instead, the joint adapter can achieve the best performance after tuning.

### D.2 ABLATION STUDY ON EVENT TIME WINDOW SIZE $N$

Robust event-to-frame conversion is an active research field in event-based vision. In this paper, we adopt the simple event histogram representation as it already gives good performance. To resist camera and object motion changes, we convert every $N$ event into one frame. The optimal $N$ varies across datasets, as they are often captured by event cameras with different resolutions. We study the effect of $N$ on few-shot classification accuracy in Fig. 6. Overall, EventCLIP is not sensitive to $N$ within a reasonable range, as the accuracy difference is smaller than 2% in most cases. One can explore better event representations to further improve EventCLIP's performance.

### D.3 SEMI-SUPERVISED LEARNING WITH EVENTCLIP

We conduct semi-supervised learning on N-ImageNet (Mini), where a few labeled data are available per class. As shown in Table 4, training on generated pseudo-labels consistently improves the accuracy compared to using only labeled data, showing a promising direction of leveraging raw data.

### D.4 N-IMAGENET PRE-TRAINED BASELINES

In our main experiments, all baselines employ backbones pre-trained on the RGB images from ImageNet. Here, we evaluate a more challenging scenario, where the backbones are *initialized from weights pre-trained on the large-scale N-ImageNet dataset*. Fig. 7 compares the few-shot accuracy of EventCLIP and the baselines. N-ImageNet pre-training provides substantial domain-specific knowledge for event-based classification, and thus greatly improves their performance. Still, EventCLIP is able to outperform the baselines with a sizeable margin when the number of data per category is small (e.g. 10-shot), and achieve competitive results with more training data. This demonstrates the effective knowledge transfer process of our approach.

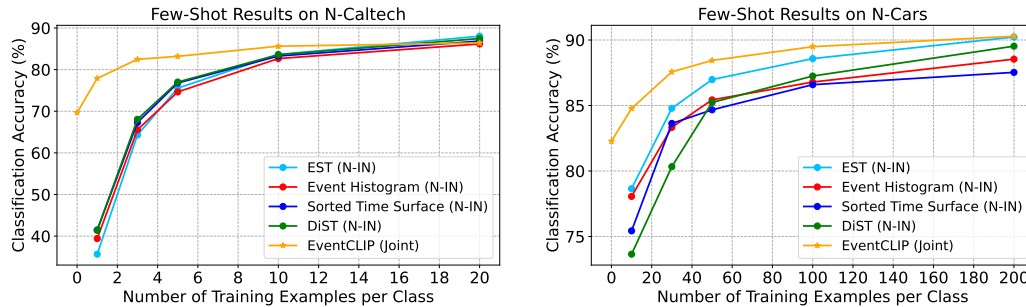

Figure 7: Comparison of few-shot accuracy between EventCLIP and baselines initialized from weights pre-trained on N-ImageNet. Our method still achieves more data-efficient learning on N-Cars, while performs competitively on N-Caltech.

## D.5 QUALITATIVE RESULTS

In Fig. 8, we visualize data from the N-ImageNet test set and its robustness variants. The original event image presents the best visual quality with both sharp object boundaries and small background noises. This is because the original data are captured under a regular camera trajectory (square) and a small moving displacement. In contrast, with an irregular moving trajectory (Variant 2, only horizontal movement), some boundaries are missing due to small image gradients along the moving direction. With a larger moving amplitude (Variant 3, 5), the event images show severe motion blur, and there are lots of background noisy events. For lighting changes, both too-low (Variant 6, 7) and too-high (Variant 9) illuminations result in distorted object boundaries. Overall, these data variations cause significant train-test discrepancy, leading to a large performance drop in event-based classifiers trained solely on event camera datasets. On the contrary, CLIP is trained on Internet-scale data covering diverse environmental conditions, which greatly improves the robustness of EventCLIP. Indeed, we successfully classify all variants of this data, while DiST fails on Variant 5, 6, and 7.

We show another example in Fig. 9. The event images under different camera motions follow similar distortions. However, under the low-light condition (Variant 6, 7), the chairs almost disappear, making the recognition task problematic. As a result, neither EventCLIP nor baselines is able to predict the correct category.

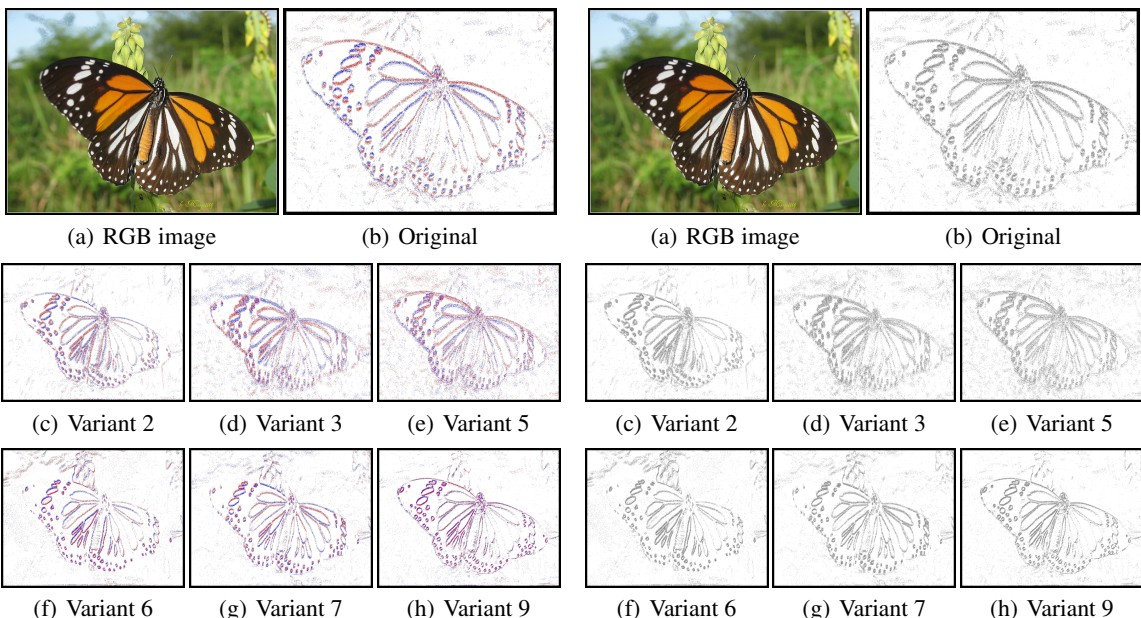

Figure 8: An N-ImageNet (Kim et al., 2021) sample from the "monarch butterfly" category. We show (a) the source RGB image from ImageNet (Deng et al., 2009) used to generate the events for reference. We plot (b) the event histograms of this data under the normal capture setting and (c-h) several robustness variants. Variant 2, 3, and 5 introduce changes in the camera trajectory, while Variant 6, 7, and 9 use different lighting during data capture. *Left*: we apply the red-blue color map for better visual quality. *Right*: the actual inputs to EventCLIP with the gray-scale color map.

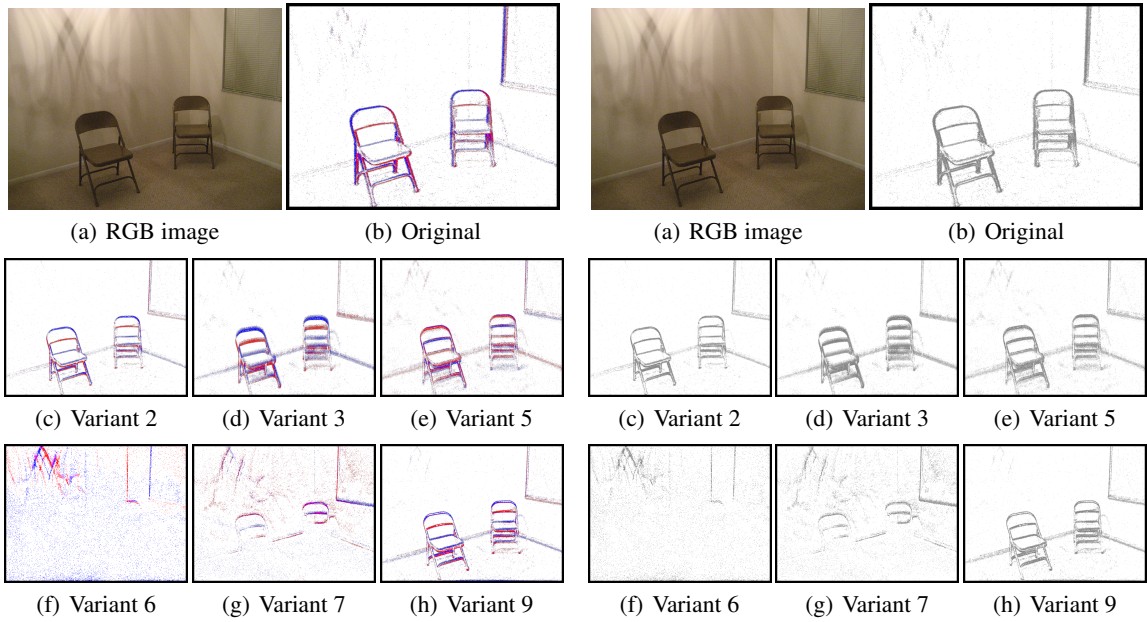

(a) RGB image     (b) Original     (a) RGB image     (b) Original

(c) Variant 2    (d) Variant 3    (e) Variant 5    (c) Variant 2    (d) Variant 3    (e) Variant 5

(f) Variant 6    (g) Variant 7    (h) Variant 9    (f) Variant 6    (g) Variant 7    (h) Variant 9

Figure 9: An N-ImageNet sample from the "folding chair" category. We again plot the red-blue colorized images at left and the gray-scale input to EventCLIP at right. Due to extreme low light, events under Variant 6 do not contain any object information, making classification impossible.

## D.6 FULL NUMERICAL RESULTS

In the main paper, we plot EventCLIP's zero-shot and few-shot classification accuracy in Fig. 3 and Fig. 4. To ease future comparison, we report all those numbers in Table 5, Table 6, and Table 7.

Table 5: Full zero-shot and few-shot classification accuracy (%) of EventCLIP on N-Caltech.

| Dataset | N-Caltech | | | | | |
|---|---|---|---|---|---|---|
| Shots | 0 | 1 | 3 | 5 | 10 | 20 |
| Visual-MLP | 69.67 | 73.52 | 77.43 | 78.55 | 82.69 | 83.28 |
| Visual-Trans. | 69.67 | 76.51 | 81.44 | 82.99 | 85.21 | 85.73 |
| Text | 69.67 | 77.35 | 81.58 | 82.81 | 84.89 | 86.33 |
| Joint | 69.67 | 77.89 | 82.48 | 83.19 | 85.62 | 86.41 |

Table 6: Full zero-shot and few-shot classification accuracy (%) of EventCLIP on N-Cars.

| Dataset | N-Cars | | | | | |
|---|---|---|---|---|---|---|
| Shots | 0 | 10 | 30 | 50 | 100 | 200 |
| Visual-Trans. | 82.28 | 84.55 | 86.62 | 87.36 | 89.51 | 90.33 |
| Text | 82.28 | 84.37 | 87.06 | 87.20 | 89.40 | 90.34 |
| Joint | 82.28 | 84.77 | 87.57 | 88.44 | 89.49 | 90.29 |

Table 7: Full zero-shot and few-shot classification accuracy (%) of EventCLIP on N-ImageNet.

| Dataset | N-ImageNet | | | | | |
|---|---|---|---|---|---|---|
| Shots | 0 | 1 | 3 | 5 | 10 | 20 |
| Visual-Trans. | 20.78 | 23.36 | 24.45 | 25.20 | 26.64 | 28.45 |
| Text | 20.78 | 24.04 | 25.47 | 26.55 | 28.24 | 30.25 |
| Joint | 20.78 | 24.25 | 25.84 | 26.96 | 28.63 | 30.57 |

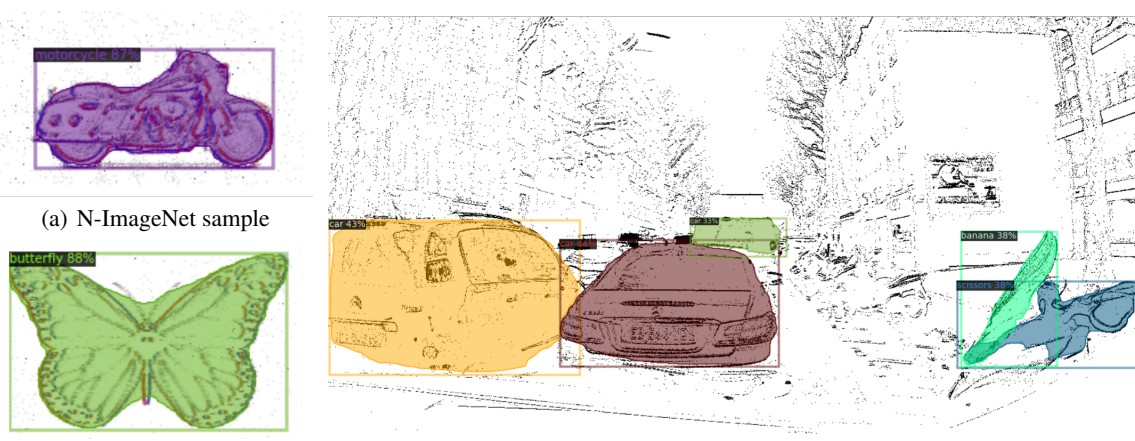

(a) N-ImageNet sample

(b) N-ImageNet sample

(c) Gen1 (De Tournemire et al., 2020) detection data

Figure 10: Transfer Detic (Zhou et al., 2022d) to perform instance segmentation on (a), (b) synthetic classification data from N-ImageNet, and (c) real-world detection data from Gen1.

## E  LIMITATIONS AND FUTURE WORKS

In this paper, we mainly focus on the event-based object recognition problem. It is still unclear how to utilize large pre-trained models for other event camera tasks, such as detection (De Tournemire et al., 2020; Perot et al., 2020) and segmentation (Wang et al., 2021a; Sun et al., 2022). We conduct preliminary experiments to transfer an open-world instance segmentation model Detic (Zhou et al., 2022d) to event data. Since Detic also builds upon CLIP, it is able to detect objects from N-ImageNet samples as shown in Fig. 10 (a), (b). However, objects present more complex motions in real-world captured events, degrading the converted event frames' visual quality drastically. Still, Detic is able to detect some objects as shown in Fig. 10 (c). But it also misses objects with sparse edges such as the motorbikes and trucks. Besides, it predicts some weird classes such as bananas and scissors due to the large domain gap. Therefore, it is worth studying better event-to-frame conversion or model adaptation methods to better leverage the pre-trained vision foundation models.

Another direction is to directly learn a joint embedding space of events and texts. Due to a lack of event-text dataset, we can leverage RGB images as the intermediate representation, as done in some recent work (Girdhar et al., 2023; Xue et al., 2023). We can first leverage the large-scale RGB video datasets such as CO3D (Reizenstein et al., 2021) to simulate event data. Then, we train an event encoder to align the extracted features with a pre-trained image encoder such as CLIP.

