# OpenReview forum: "EventCLIP: Adapting CLIP for Event-based Object Recognition"
_ICLR.cc/2024/Conference — ICLR 2024 Conference Withdrawn Submission_

### Official Review · Reviewer_4a2f · 2023-10-26

**Soundness:** 3 good
**Presentation:** 3 good
**Contribution:** 3 good
**Rating:** 6
**Confidence:** 4

**Summary:**

This work study the adaptation of foundation model for zero-shot and few-shot object recognition for event camera. The results suggest vision-language foundation models can be used for Event Camera, which can open up practical use cases for event camera in real-world scenario.

**Strengths:**

+ This work explores how can foundation model be applied for Event Camera's data for zero- or few-shot object recognition task. A Transformer-based adapter is proposed to aggregate temporal information over event frames (edge maps).
+ To improve the performance on unseen object, a method is proposed to learn EventCLIP from unlabeled data where pseudo labels is obtained via test-time-augmentation.
+ Experiment results also suggest the 2D VLMs knowledge complements the event-camera data trained models.
+ Experiment results show significant improvement over the compared baseline.

**Weaknesses:**

- It would be better if more VLMs can be adapted for the proposed method to show its generalizability to more VLMs.
- The key contribution of this work is successful adapt VLMs for Event Camera based object recognition. However, the technical contributions is considered relatively low as most approaches are studied in prior work.

**Questions:**

- The domain-specific event knowledge and the 2D pre-trained knowledge are shown to be complement each other. On the other hand, would there be cases where the errors from both knowledge source actually enhance the errors in the final prediction? Has the author encounter such cases.

---

### Official Review · Reviewer_rcqs · 2023-10-29

**Soundness:** 3 good
**Presentation:** 3 good
**Contribution:** 2 fair
**Rating:** 3
**Confidence:** 4

**Summary:**

This article primarily introduces EventCLIP, which applies CLIP to event data. The main work includes two parts: (1) This method first extends the image encoder of CLIP to event data by converting the raw events into a two-dimensional grid-based representation. The authors divide the event stream into multiple time windows and convert each window into 2D frames. Following the approach of Radford et al. (2021), they construct text prompts by placing class names into manually crafted templates and extracting text features as weights for zero-shot classifiers. (2) To further improve performance, this method proposes a feature adapter to aggregate temporal information on event frames and enhance text embeddings for better alignment with visual inputs. State-of-the-art performance is achieved in few-shot scenarios. When fine-tuning the entire dataset, our method outperforms all existing event classifiers. Additionally, we explore the practical applications of EventCLIP, including powerful event classification and unsupervised event recognition, where our method surpasses previous benchmarks specifically designed for these tasks.

**Strengths:**

+ This article is the first to apply CLIP to event data. It proposes the conversion of raw event data into a two-dimensional grid-based representation, enabling direct compatibility with CLIP models.
+ The proposed method has achieved good results on existing event-based datasets.

**Weaknesses:**

+ I have doubts about the novelty of this paper. It seems more like the application of CLIP technology to a niche data type, event data, and the methods involved in it have already appeared in related research in the past two years.
+ This method first compresses the time window of event data to form a single-frame representation. This idea can be found in video-based CLIP applications, such as "ActionCLIP: A New Paradigm for Video Action Recognition" by Mengmeng Wang, Jiazheng Xing, Yong Liu in 2021. However, in general, the action recognition task involves feature-level fusion, while this method operates at the input level.
+ The adapter part of this method is also quite straightforward, mainly utilizing fine-tuning techniques and their variations.

**Questions:**

The description of this paper is relatively clear to me. I'm not quite sure where the novelty lies in the transformation of the input and why CLIP can directly work on the transformed data.

Translation:
I find the description of this article relatively clear. I am not entirely certain about the novelty of the transformation approach used for the input and why CLIP can effectively handle the transformed data directly.

---

### Official Review · Reviewer_FLiL · 2023-10-31

**Soundness:** 3 good
**Presentation:** 3 good
**Contribution:** 2 fair
**Rating:** 5
**Confidence:** 2

**Summary:**

This paper propose to use CLIP classifier to event-based images. The authors propose to render the event histograms as 2-channel images (stacked temporally), and feed each image to a pretrained CLIP classifier. The authors studied how to render the event-image (gray/ RB/ learned), how to aggregate temporal features, etc. Results on 3 event-classification datasets show the proposed method achieves state-of-the-art performance on both fewshot and finetuning setting.

**Strengths:**

- The task of adapting vision models for even cameras are important. This paper takes a good step in this direction.

- The results are strong. The proposed method achieved the state-of-the-art performance on multiple settings (few-shot, finetuning) and multiple datasets (N-Caltech, N-ImageNet, N-Cars).

- The paper is well written, well-motivated, and easy to follow.

**Weaknesses:**

- My main complaint is that I see no "surprise" in this paper. CLIP is a well known strong model for zero-shot classification, and it is expected that using the image and the text embedding of CLIP can outperforms counterparts that didn't use CLIP.

- I feel the most "novel" part of this project is to render an event-based image as a 2 channel gray image. First, did any existing paper did that? The related work section mentioned some, but please elaborate more how the rendering technique used in this paper is different from them, and ideally compare if the proposed one is optimally.

- If "rendering event-based image into regular image" is a key contribution here, I would be curious if the observation hold for other tasks, e.g., standard image classification using initialization from standard imagenet pretrained checkpoint. Note this is NOT a required experiment in the rebuttal.

- The technical designs, e.g., the feature adapter, ensembling with a traditional model, are straightforward, and I don't feel we can claim good novelty here.

**Questions:**

Overall this paper has good results on an important task. My first impression of the paper is it just takes a well-known working idea from images to event-images, and thus not feeling it is challenging or novel. In the rebuttal, I hope to know 1. how the proposed event-to-regular image technique is different from existing works and 2. If it is different, can it also works with other image-pretrained weights other than CLIP. My current rating is a weak reject.

---

### Official Review · Reviewer_KXch · 2023-11-01

**Soundness:** 2 fair
**Presentation:** 1 poor
**Contribution:** 1 poor
**Rating:** 3
**Confidence:** 4

**Summary:**

EventCLIP, a novel approach that utilizes CLIP for zero-shot and few-shot event-based object recognition. CLIP’s image encoder is generalized to event data by converting raw events to 2D grid-based representations. To further enhance performance of the zero shot setting, a few shot setting is proposed with the availability of some class labels. Moreover, EventCLIP can be employed to improve the robustness of existing classifiers via model ensemble, or learn from unlabeled data with self-training.

**Strengths:**

•	All the necessary background information needed to understand the proposed approach is presented in the paper, hence there is no requirement of a lot of prior knowledge of the reader.

**Weaknesses:**

•	The novelty of the paper is questionable in terms of the zero-shot setting, since in that case the only novelty is the event to frame conversion and then feeding those frames to a pre-trained CLIP model (section 3.2). This conversion method is pre-existing (event histogram) as well.

•	Even in the few shot settings, although the image feature adapter module is a new addition to the CLIP model, it is inspired from previous work (PointCLIP CVPR23 [1]) and thus again defeats the purpose of novelty (section 3.3). There is no comparison shown with PointCLIP

•	The paper is not well organized. Have to move back and forth between the main paper and appendix to see parts of the results, which should have been there in the experiments section

**Questions:**

•	Fig 1. says existing event-based models are trained from scratch on closed-set datasets. They fail on events with unseen categories, camera motions or lighting. PointCLIPv2 (ECCV23) [2] solves this exact problem in open world setting. However, the proposed approach also lacks the ability to generate accurate recognition in low-light situation, so does not really solve the generalization issue in terms of lighting variation as they claim.

•	Section 3.1 should go in the related work and not methodology

•	Section 4.2 Results show effectiveness of the approach based on the accuracy reported in Table 1a, however, without comparing the approach with other methods, cannot comment on effectiveness of the approach. What was the data variation of this table? original or trajectory/brightness variations/average of all? This section needs rewording.

•	Figure 6 is the ablation study on which dataset? Why was the ablation study not consistent over datasets? Different experiments seem to have been done on different datasets (Table 1b is on N-Caltech, N-ImageNet, whereas Table 1c is on N-Caltech and N-Cars)

•	Report results for section 4.5 robust event classification on the proposed approach itself without any ensemble of models in both few shot and zero shot settings across all data variations.

•	Section 4.6 says fully unsupervised learning is done and it is better than one of a previous work (Cho et al., 2023), however, in the unsupervised setting the results are not actually better (27.08 % vs 31.28%).  The results get better in the semi-supervised setting, however, since these two are two different learning approaches, the results cannot be comparable. Need clarification of this contradiction.

---

### Official Review · Reviewer_rCwv · 2023-11-05

**Soundness:** 3 good
**Presentation:** 2 fair
**Contribution:** 2 fair
**Rating:** 3
**Confidence:** 5

**Summary:**

This paper develops a new EventCLIP based on CLIP for zero/few-shot event recognition. In practice, the feature adaptors are used to fuse image features and refine text features to enhance the performance. To support it, extensive experiments are conducted on various datasets, demonstrating that the proposed method outperforms state-of-the-arts.

**Strengths:**

1. The authors successfully applied 2D CLIP model on event based object recognition by integrating image features in consecutive frames.
2. The comprehensive experiments show the effectiveness of the proposed method. The authors explored the possibility of learning from unlabeled data with lightweight adaptors.

**Weaknesses:**

1. This paper extended the original CLIP to temporal EventCLIP by adding both image and text feature adapters. However, the two adapters come from previous works in a pretty straight-forward manner. That is, the novelty of this paper is limited. The authors should clarify the scientific contribution of this paper.
2. The authors claimed that the final prediction of EventCLIP should be order-invariant. In my opinion, some events are naturally order sensitive. For example, standing up is the reverse of sitting down. Then how does the proposed model distinguish these events?
3. In Figure 2, the proposed event histograms are not clear. What is the insight of this operation?
4. The authors stated that the training data is so limited that the CLIP backbone cannot be well finetuned in Sec. 3.3. On the contrary, the accuracy is improved by finetuning the model in Sec. 4.4.
5. Many details are explained in the appendix, some of which should be put in the main text.

**Questions:**

Please see my above comments. The authors should highlight the novel scientific contributions of this paper.

---

### Official Review · Reviewer_RBgA · 2023-11-07

**Soundness:** 3 good
**Presentation:** 3 good
**Contribution:** 3 good
**Rating:** 3
**Confidence:** 3

**Summary:**

In their study, the authors introduce EventCLIP, a novel approach that utilizes CLIP for zero-shot and few-shot event-based object recognition. They start by adapting CLIP's image encoder to handle event data by transforming raw events into 2D grid-based representations. To further enhance performance, they introduce a feature adapter that aggregates temporal information over event frames and refines text embeddings to better align with the visual inputs.

**Strengths:**

1. This is the first work to leverage CLIP to  event-based object recognition.
2. The paper is well-written and easy to follow.
3. The experimental results are good.

**Weaknesses:**

This article directly introduces CLIP without explaining its significance in this field. My concerns are as follows:
1. What is the key problem in event-based object detection ?
2. Can CLIP truly address some key points of this domain?
3. Did you try other foundation models ?

**Questions:**

See the weakness. If you address my concerns, I will improve my score.